# Validation of the Portuguese Version of the Healthy Lifestyle Questionnaire

**DOI:** 10.3390/ijerph17041458

**Published:** 2020-02-24

**Authors:** Marco Batista, Marta Leyton-Román, Samuel Honório, Jorge Santos, Ruth Jiménez-Castuera

**Affiliations:** 1Department of Sports and Well-being, Instituto Politécnico de Castelo Branco, Rua Prof. Dr. Faria de Vasconcelos, Higher School of Education—SHERU (Sport, Health and Exercise Research Unit), 6200 Castelo Branco, Portugal; marco.batista@ipcb.pt (M.B.); samuelhonorio@ipcb.pt (S.H.); jorgesantos@ipcb.pt (J.S.); 2Sport Studies Centre, Rey Juan Carlos University, Pso Artilleros, s/n, 28032 Madrid, Spain; marta.leyton@urjc.es; 3Didactic and Behavioural Analysis in Sport Research Group, Faculty of Sport Sciences, University of Extremadura, Avda. de la Universidad, s/n, 10003 Cáceres, Spain

**Keywords:** healthy lifestyles, confirmatory factorial analysis, sport, veterans, psychometrics

## Abstract

The main objective of this study was the validation of the Healthy Lifestyle Questionnaire (EVS II), using a confirmatory factorial analysis of the measurement model, with veteran athletes. A total of 348 veteran Portuguese athletes of both genders, aged between 30 and 60 years (*M* = 41.64, *SD* = 9.83), of whom 200 were males and 148 were females, from several sports. The results of the confirmatory factor analysis demonstrate the adequacy of the adapted version of the EVS II, as the factorial structure (6 factors/24 items) has acceptable validity indexes: χ^2^ = 305.925, *p* = 0.000, d*f* = 120.017, χ^2^*/df* = 2.549, *NFI* (Normed Fit Index) = 0.909, *TLI* (Tucker Lewis Index) = 0.918, *CFI* (Comparative Fit Index) = 0.944, *GFI* (Goodness of Fit Index) = 0.944, *AGFI* (Adjusted Goodness of Fit Index) = 0.909, *SRMR* (Standardized Root Mean Square Residual) = 0.048, *RMSEA* (Root Mean Square of Approximation) = 0.060, allowing evaluation of the dimensions of balanced diet, respect for mealtime, tobacco and alcohol consumption, other drugs consumption and resting habits. The adaptation to sport of the Portuguese version of EVS II can be used with reasonable confidence in the evaluation of healthy lifestyles in the context of sport.

## 1. Introduction

Healthy lifestyles are an obstacle to diseases and include preventive health, good nutrition, weight control, leisure, regular physical activity, periods of rest and relaxation, the capacity to face adverse conditions or situations and establishment of affective relationships of solidarity and citizens, adopting a posture of being in the world with the purpose of living with quality [1].

The study of healthy lifestyles can be marked up to the present time by three great periods: a first period beginning in the nineteenth century until the middle of the twentieth century; a second period, which ends the second half of the twentieth century and a third period that focuses on the present [2]. The same authors state that during the first period, the studies emphasize healthy lifestyles, dependent on a sociological vision and individual factors, of the individuals who belonged to a certain social stratum and could keep that. In the second period, there were studies that reported an incorporation of healthy lifestyles in the field of health and studies with isolated conducts, apparently salutary or conducive to a healthy living style. In the third period, there was a development up to the present day, in which work reflects healthy behaviours integrated in different contexts, or even the relationship of different dimensions with healthy lifestyles, such as motivation or basic psychological needs, seeking to perceive self-determination for a particular practice and the adoption of healthy behaviours.

It should be noted that, only in the second and third chronological periods of the study of lifestyles, did this construct begin to be evaluated, according to valid and reliable instruments for this purpose. Examples of these instruments are those that were developed and applied in the works of several authors [3,4,5,6,7,8,9,10,11,12,13].

The EVS instrument is derived from The Health Behavior in School Children (HBSC) [6]. This instrument initially evaluated health behaviours in the school population, which determines sociodemographic variables, healthy lifestyle variables, daily life variables and psychosocial health variables. This instrument reflects an inventory of health behaviours in students [3], with a factorial distribution of 29 items [7]. 

This instrument was later adapted and used in Spain [7]. The Healthy Lifestyle Questionnaire (EVS) proposes an instrument of 30 items, distributed by six factors that determine behaviours, more specifically, eating habits and the subfactors: balanced diet and respect for mealtime, resting habits, tobacco and alcohol consumption and other drugs consumption. In the same line of research [14], the relationship between physical activity and sport on behaviours belonging to healthy lifestyles was analyzed, using a sample of 402 students of Physical Education, from 14 to 18 years of age. The results showed that this practice positively and significantly predicts the balanced diet and respect for mealtime and, without showing statistical significance, resting habits. Some authors [15,16] demonstrated the relationship between the practice of physical-sports activities and the most self-determined motivation [17], with the consequence of maintaining healthy lifestyles [18,19]. It is also important to note that some authors [20] indicated that tobacco consumption was a significantly predisposing factor for alcohol consumption.

The preliminary evaluation of EVS for the Portuguese language [21] was developed with a sample of Portuguese secondary school students, where goodness indexes of the confirmatory factorial analysis were acceptable. The structure of the EVS was composed of 26 items distributed by five factors: tobacco and alcohol consumption, other drugs consumption, respect for mealtimes, and maintenance of a balanced diet. The resting habits factor was eliminated because its items presented a factorial weight of 40.

Later, other authors [22] sought to adapt to the context of the Portuguese veteran sport of EVS. The overall results of the model indicated a good fit, as well as a good validity concurrent with the motivation evidenced for the practice. The structure of the EVS was composed of 20 items distributed by three factors: tobacco consumption, resting habits and eating habits. This version of EVS did not include all the original dimensions of the questionnaire, specifically alcohol consumption and consumption of other drugs, since they were not considered adequate for the evaluation of the collective veteran sports. There were also problems with the factorial load of some items related to the dimensions of balanced diet and respect for the mealtimes, and the grouping of the same ones was chosen, being denominated by the factor of eating habits.

The Spanish validation of EVS [2] was developed with a sample of 812 participants between the ages of 14 and 88 years. The overall results of the model indicated an optimal fit, as well as a good concurrency against the motivational orientation based on the theory of self-determination. The structure of the EVS was composed of 12 items divided by four factors: tobacco consumption, resting habits, respect for mealtime, and maintenance of a balanced diet. Currently, the healthy lifestyle construct has a close relationship with the modern food pyramid and is investigated according to multivariate models, which involve quality of life or absence of disease.

The present study aims to validate the Healthy Lifestyle Questionnaire (EVS II) for the Portuguese language.

The use of this questionnaire will allow a more comprehensive characterization of the healthy habits and lifestyles of athletes, regarding the validations of the EVS [21,22], supporting translation of the Spanish version [2] to Portuguese. The Healthy Lifestyles Questionnaire (EVS II) itself presents as a new and a more appropriate instrument, which has an adequate number of items per factor.

## 2. Materials and Methods

### 2.1. Research Design

This study focuses on quantitative empirical studies and refers to a descriptive study of populations through surveys [23], in this case, veteran athletes. There was no manipulation of the independent variables, that is, the variables are presented as they are, without any interference by the researchers [23].

### 2.2. Participants

The study sample consisted of 348 veteran Portuguese athletes of both genders, aged between 30 and 60 years (*M* = 41.64 *SD* = 9.83), of whom 200 were male with *M* = 43.82, *SD* = 8.62 years and 148 were female with *M* = 40.26, *SD* = 9.96 years. Athletes were from several sports, with continuous practice, of at least, two years. These athletes register mostly between 3 to 5 hours of weekly training, with *M* = 19.5, *SD* = 12.2 years of practice. Data are from practitioners of team sports, such as football, roller hockey, rugby, basketball, as well as individual sports, namely, tennis, judo, athletics, mountain biking, triathlon, horse riding or cycling. The type of sampling used for sample selection in the present study was intentional non-probabilistic [23], and, as it was not based on a probabilistic basis, an intentional approach was done to subjects with certain specific characteristics.

### 2.3. Instruments

The Healthy Lifestyle Questionnaire (EVS II) was used, which was translated and adapted into Portuguese from the Spanish version of the Healthy Lifestyle Questionnaire (EVS) [2]. The EVS II presents a 30-item version, using a Likert-type scale that varies between fully disagree (1) and fully agree (5). The questionnaire was divided into the areas of eating habits, consumption of harmful substances and resting habits. These were grouped into six factors: dietary habits that include (a) a balanced diet and (b) respect for mealtime, consumption of harmful substances that integrate (c) tobacco consumption, (d) alcohol consumption, (e) consumption of other drugs, and resting habits with a single designated dimension, (f) resting habits. The measurement of eating habits included 11 items, namely balanced diet (e.g., “normally, as fish two or more times per week.”) had 6 items, and respect for meal times (e.g., “I usually respect meal times.”) had 5 items. The consumption of harmful substances had 15 items; tobacco consumption (e.g., “I smoke in the usual way”) had 5 items, alcohol consumption (e.g., “I drink alcohol on a regular basis at weekends, liqueurs, wines, beverages...”) had 5 items, and consumption of other drugs (e.g., “I have tried drugs (charros, marijuana, cocaine, stimulants,...”)) had 5 items. Resting habits had 4 items (e.g., “I usually sleep 7–8 hours daily”). 

The Behavioral Regulation in Sport Questionnaire (BRSQ) [24] is composed of 24 items, divided into 6 subscales evaluated according to a 7-level Likert scale, ranging from 1 (totally disagree) to 7 (totally agree). The six subscales allow one to determine the variables of amotivation, external motivation, introjected motivation, identified motivation, integrated motivation and intrinsic motivation.

These items also reflect the types of motivation underlying the motivational continuum of the self-determination theory, namely amotivation, controlled motivation (which aggregates external motivation and introjected motivation), and autonomous motivation (which aggregates identified motivation, integrated motivation and intrinsic motivation). For this theoretical conceptualization, BRSQ can be used to determine only three variable variables, which was our option in the present study, to determine the levels of amotivation, controlled motivation and autonomous motivation. This scale was validated for the sport context in the Portuguese language [25].

### 2.4. Procedures

The study was approved by the Bioethics and Biosafety Commission of the University of Extremadura (Spain) under the registration number R011-0322020, following the guidelines of the Declaration of Helsinki. All participants were treated according to the ethical guidelines of the American Psychological Association with respect to participants’ consent, confidentiality, and anonymity. Informed written consent was obtained from all participants. 

An inverse translation was performed [26] of the Healthy Lifestyles (EVS) items, first translated into Portuguese and later translated again by a translator from the research group for the Spanish language, where he observed a great similarity also with the original questionnaire in English after the retroversion process. Next, the items were evaluated by three experts in the field who considered that they were adequate to evaluate the construct for which it was created. Once translated, the questionnaire was administered to a small group of practitioners of similar ages to the final sample of the study to verify their correct understanding, where they did not notify any problems of reading comprehension.

In a subsequent phase, selection of sports centers, such as clubs and associations, was carried out, serving as a convenience sample [23]. For the collection of information, we contacted the veteran athletes directly to request their collaboration in the requested study. After their agreement, they signed an informed consent. The administration of the definitive questionnaire of the Healthy Lifestyle Questionnaire (EVS II) was carried out in the presence of the principal investigator, to briefly explain the objectives and structure, as well as the filling requirements. During the filling process, the main investigator was available for any problem that might arise. The time for filling was approximately fifteen minutes.

### 2.5. Data Analysis

The statistical analysis of the data was done through the statistical software SPSS (version 23.0 for Windows, SPSS, Inc., Chicago, IL, USA). Firstly, we filtered the data to make sure there was no missing data. Then, having no missing data, we observed the existence of normal data obtained. For the univariate analyses of normality, the asymmetry and kurtosis indicators of each item that composed the EVS II were used first. The author [27] also proposed the limits, in absolute value, and considered values up to 2 for asymmetry and 7 for kurtosis for a behaviour similar to normal; between 2 and 3 for asymmetry and between 7 and 21 for kurtosis for moderately normal behaviour; and values greater than 7 in asymmetry and 21 in kurtosis for extremely normal behaviour.

To confirm the structure of the respective factors with their corresponding items, confirmatory factorial analysis (CFA) was carried out using the software EQS (version 6.1 for Windows, Multivariate Software, Inc., Los Angeles, IL, USA). To evaluate the adequacy of our given structural equation model, the maximum likelihood method (ML) was used. In the analysis, a combination of indexes [28] was used, and for this reason, the indicators recommended were: χ^2^, χ^2^/df, NFI (Normed Fit Index), TLI (Tucker Lewis Index), CFI (Comparative Fit Index), GFI (Goodness of Fit Index), AGFI (Adjusted Goodness of Fit Index), RMSEA (Root Mean Square of Approximation) and SRMR (Standardized Root Mean Square Residual) [28].

The χ^2^ indicates a similarity of the covariates observed with those that are predicted in the hypothetical model, with values for a good fit 0 ≤ χ^2^ ≤ 2df and an acceptable fit 2df < χ^2^ ≤3df. However, it is very sensitive to the size of the sample, so it is recommended to complete with χ^2^/df, whose values below 2 indicate a very good fit of the model, although values below 3 are considered acceptable [28].

The incremental indexes (NFI, TLI, CFI, GFI and AGFI) compare the hypothetical model with the null model and are not affected by the sample size. Values greater than 0.90 are considered acceptable, but if they are greater than 0.95 they are considered good [28].

The RMSEA and SRMR error rates should be less than 0.08 for an acceptable fit and should be less than 0.05 for a good fit [28], and the standardized factor loads should all be statistically significant (*p* < 0.01) [28]. 

In the determination of the model, we investigated the multivariate normal distribution criterion, using the normalized Mardia coefficient, which should be less than 5, allowing the estimation of structural models using the maximum likelihood method [28]. In the case of non-compliance with the normal multivariate distribution, we followed the recommendations for using the robust maximum likelihood estimation method [28] with the application of the statistical corrective measure of robustness χ^2^ [29].

Later, in order to perform the confirmatory factor analysis (CFA), the construct validity was estimated, respecting the criterion of eliminating those items whose regression weight does not have an adequate value (greater than 0.40), and the factorial loads of each item should be significant. We also determined the internal consistency of each of the factors resulting from the factorial analysis Omega coefficient (ω), which expresses that the coefficient of reliability must be above 0.70 [30].

Composite reliability (degree of consistency between latent construct indicators) and mean extracted variance (the amount of variance of indicators, captured by the construct, compared with that obtained by the measurement error) was estimated. Given the composite reliability, the minimum level is 0.70, and the mean variance extracted should be greater than 0.50, to conclude that a substantial amount of the variance is captured by the construct [30].

In order to verify whether the number of factors is reasonable based on the specific measurement model presented, the authors [30] presented a very simple approach for this purpose, suggesting the calculation of the Omega hierarchical subscale coefficient (OmegaHS). Consequently, OmegaHS may be viewed as a representation of effect size that is essentially unaffected by sample size. The same authors [30] suggest that OmegaHS can be considered as an indicator of latent variable strength specific to the factors that constitute a variable. Values close to 0.00 are indicative of a very weak specific latent variable, while values close to 1.0 are indicative of a very strong specific latent variable. They propose that relatively small, typical, and relatively large OmegaHS values correspond to the following guidelines: relatively small <0.20; typical 0.20 to 0.30; and relatively large >0.30. OmegaHS values <0.10 should probably be considered as relatively very small.

A descriptive analysis was carried out by the determination of means and standard deviations of each extracted factor, and the concurrent validity was evaluated through a bivariate correlation analysis. This assessment of concurrent validity is justified because, according to the theoretical conceptual framework of the theory of self-determination [17], autonomous motivation appears positively correlated with healthy behaviours and negatively correlated with unhealthy behaviours. Instead of this logic, it is natural for controlled motivation and motivation to appear negatively correlated with healthy behaviours and positively with non-healthy behaviours [18,19].

## 3. Results

### 3.1. Confirmatory Factor Analysis

In the present work, we have chosen to eliminate six items, since they did not meet the factorial load equal to or greater than 0.40, as proposed by the author [31]. In the balanced diet factor, item 12 (Like sweets, cakes, ... at most once or twice a week) was eliminated, in the factor of tobacco consumption, item 5 [I consider that tobacco helps to relate to me, (alcohol makes it better)] and item 16 (I have the feeling that I’m always drinking more alcohol) were eliminated, in the dimension of other drugs consumption, item 3 (I usually take some drug) was eliminated, and in the resting habits dimension, item 22 (I usually take a nap for approximately 30 min) was eliminated.

Confirmatory factor analysis to evaluate the six-factor model of the Healthy Lifestyle Questionnaire-EVS II, showed that the 24 items were grouped into six factors, respectively: balanced diet (5 items), respect for mealtime (5 items), tobacco consumption (4 items), alcohol consumption (3 items), other drugs consumption (4 items) and resting habits (3 items). 

Likewise, the standardized factor loads were all statistically significant (*p* < 0.01), so it can be concluded that the model presented in Figure 1, at the analytical level, presents satisfactory results.

After a first analysis, the general results of the model indicated a reasonable fit of the adapted version of the Healthy Lifestyle Questionnaire (EVS II), which was composed of 24 items: *χ^2^* = 305.925, *p* = 0.000, *df* = 120.017, *χ^2^/df* = 2.549, *NFI* = 0.909, *TLI* = 0.918, *CFI* = 0.944, *GFI* = 0.944, *AGFI* = 0.909, *SRMR* = 0.048, and *RMSEA* = 0.060. With these results, the structural model reveals a satisfactory overall fit, having models with satisfactory adjustment in previous versions, although with a smaller number of analysis dimensions than the EVS II.

### 3.2. Analysis of Internal Consistency and Convergent Validity

Table 1 shows the internal consistency values of EVS II. The internal consistency of each of the factors resulting from the factorial analysis (McDonald’s Omega ω), presented the following results: 0.81 balanced diet, 0.90 respect for mealtime, 0.96 tobacco consumption, 0.76, alcohol consumption, 0.78 consumption of other drugs and 0.74 resting habits. The mean extracted variance and the composite reliability for each factor were 0.82 and 0.53 in the balanced diet, 0.91 and 0.71 for respect for mealtime, 0.96 and 0.85 for tobacco consumption, 0.79 and 0.51 for alcohol consumption, 0.78 and 0.51 in the consumption of other drugs, and 0.77 and 0.54 in resting habits, fulfilling all the factors evaluated in the assumptions [30].

To verify whether the number of factors is reasonable based on the specific measurement model, according to the equations proposed by the author [30], OmegaHS values were obtained to balance diet by 0.27, respect for mealtime 0.36, tobacco consumption 0.49, alcohol consumption 0.35, other drugs consumption 0.27 and resting habits 0.34, these values being typical or relatively large.

From a descriptive point of view (Table 2), the values of healthy lifestyles were obtained in this group of veteran athletes, with higher means in balanced eating behaviours ($\bar{x}$ = 3.63 ± 0.80), respect for mealtime ($\bar{x}$ = 3.64 ± 0.89), and lower means in smoking average ($\bar{x}$ = 1.57 ± 1.02), alcohol consumption ($\bar{x}$ = 1.52 ± 0.68) and consumption of other drugs ($\bar{x}$ = 1.55 ± 0.74). In the motivation variables, the veteran athletes demonstrated a high autonomic motivation ($\bar{x}$ = 5.43 ± 0.89), and reduced values of controlled motivation ($\bar{x}$ = 1.99 ± 1.07) and amotivation ($\bar{x}$ = 2.02 ± 1.20).

The evaluation of concurrent validity through a bivariate correlation analysis, most of the correlations between the variables of the EVS II and the BRSQ were significant and in the expected direction. Autonomic motivation is positively correlated with a balanced diet, respect for mealtime and resting habits. It assumes a negative correlation like tobacco consumption, alcohol consumption and consumption of other drugs. Controlled motivation and amotivation assumed correlations in the opposite direction to autonomous motivation, given the different variables of healthy lifestyles.

## 4. Discussion

The main objective of the present study was to broaden previous research on healthy lifestyles, particularly associated with the practice of veteran athletes, through the validation of the Healthy Lifestyle Questionnaire—EVS II, for the Portuguese competitive sport context. 

According to that, each new application of a measuring instrument represents a contribution to improving the theoretical value of the research domain [27]. This study extends this core of knowledge, confirming the validity of the EVS II instrument in a research context, as well as through improved knowledge of how to help sports and exercise psychologists, to understand healthy practices and health indicators in veteran athletes.

Measuring the internal consistency of each of the calculated factors, using McDonald’s omega, we obtained values greater than or equal to 0.70 in both measures, as proposed by the authors [30].

We estimated the composite reliability, the mean variance extracted and OmegaHS for each factor, and we observed that the values obtained correspond to the indicators proposed [30], to conclude that a substantial amount of the variance is captured by the construct, where the composite reliability must present a minimum value of 0.70, the mean variance extracted is greater than 0.50, and OmegaHS sets to typical centered values 0.20 to 0.30 and relatively large values > 0.30.

Confirmatory factor analysis showed that the 24 items were grouped into six factors, respectively: balanced diet (5 items), respect for mealtimes (5 items), tobacco consumption (4 items), alcohol consumption (3 items), consumption of other drugs (4 items) and resting habits (3 items). With these results, the structural model reveals a satisfactory overall fit, having models with satisfactory adjustment in previous versions, although with smaller number of analysis dimensions than the EVS II.

The results obtained through the psychometric quality indexes revealed [28] an acceptable fit in *χ^2^*, and in the value of *χ^2^/df*, NFI, GFI and RMSEA. They showed good fit in the AGFI and SRMR indexes. Despite the values of TLI and CFI being very close to those indicated [30], they did not fail to comply with the values proposed by some authors [32,33].

These results are consistent with previous research using EVS [2,21,22] and confirm the importance of each of the six dimensions in understanding the healthy lifestyles of athletes. If we observe the results of the preliminary validation of EVS [21], and the work of other authors [2,22], with those obtained in our study, both presented good psychometric properties, based on what the literature advises [28,32,33], with EVS appearing in these four studies as a reliable tool for assessing healthy lifestyles. However, we emphasize that the validation of the EVS II is the adapted version that best respects the initial model of the questionnaire presented by another author [6] with six factors extracted.

The previously validated versions found problems with some items with a factorial load lower than 0.40 [31], leading to the elimination of some factors extracted, such as the resting habits, in the preliminary research [21], or agglutination of items of the dimensions of balanced diet and respect for mealtime, which in the Portuguese research [22] gave rise to the size of eating habits. In the present research, we have also opted to eliminate some items, since they did not meet the factorial load equal to or greater than 0.40, as proposed by the author [31], and six items were eliminated.

This instrument has the potential to become more refined as new contributions on healthy lifestyles arise, where in later studies it will be interesting and very convenient for these items to be remeasured and tested in order to obtain other models of valid equations, such as that proposed by the authors [28].

In the descriptive analysis, the results showed that the participants in the study tend to value the items of the questionnaire, which in fact seems to be demonstrated by the moderate and high averages in the dimensions of balanced feeding, respect for mealtime and resting habits, as well as reduced means in the dimensions of tobacco consumption, alcohol consumption and consumption of other drugs. This thus shows the theoretical importance underlying the construction of healthy lifestyles. The same descriptive trend was obtained in some previous studies [2,21,22]. Furthermore, the results of these studies supported concurrent validity through the analysis of bivariate correlations. In the present study, most of the correlations between the EVS II and BRSQ variables showed significant associations, emphasizing the validity in this work, particularly with the motivation continuum [17]. The authors [34], in a meta-analysis with most of the non-experimental studies, determined that there is a strong relationship between the self-determination theory [17] and positive health behaviours. Analogous observations were obtained in several studies [14,18,19,20,35]. 

In future applications, it would be interesting to observe additional samples with athletes from other countries, and data could also be collected to compare the validity of the scale in different cultural contexts. We also point out the interest of the possible inclusion of the practical factor of physical exercise in the questionnaire, which in future studies can be measured in the same way as other research has already done, with other specific instruments. It may be interesting to apply studies that are based on the trans-theoretical model of motivation, or on the theory of planned behaviour or even others that are based equally on the theory of self-determination [17] and to evaluate the adoption of healthy lifestyles in different strata of the population that show the practice of physical or sports activity.

## 5. Conclusions

With this study, it is our understanding that the Portuguese version of the Healthy Lifestyle Questionnaire (EVS II), with six factors, can be used with confidence in the evaluation of healthy lifestyles, underlying behaviour of eating habits, consumption of harmful substances and resting habits, in the context of sport. The results indicate that the factorial and reliability validity of the Portuguese version of the Healthy Lifestyle Questionnaire (EVS II) is acceptable for the sports field.

## Figures and Tables

**Figure 1 ijerph-17-01458-f001:**
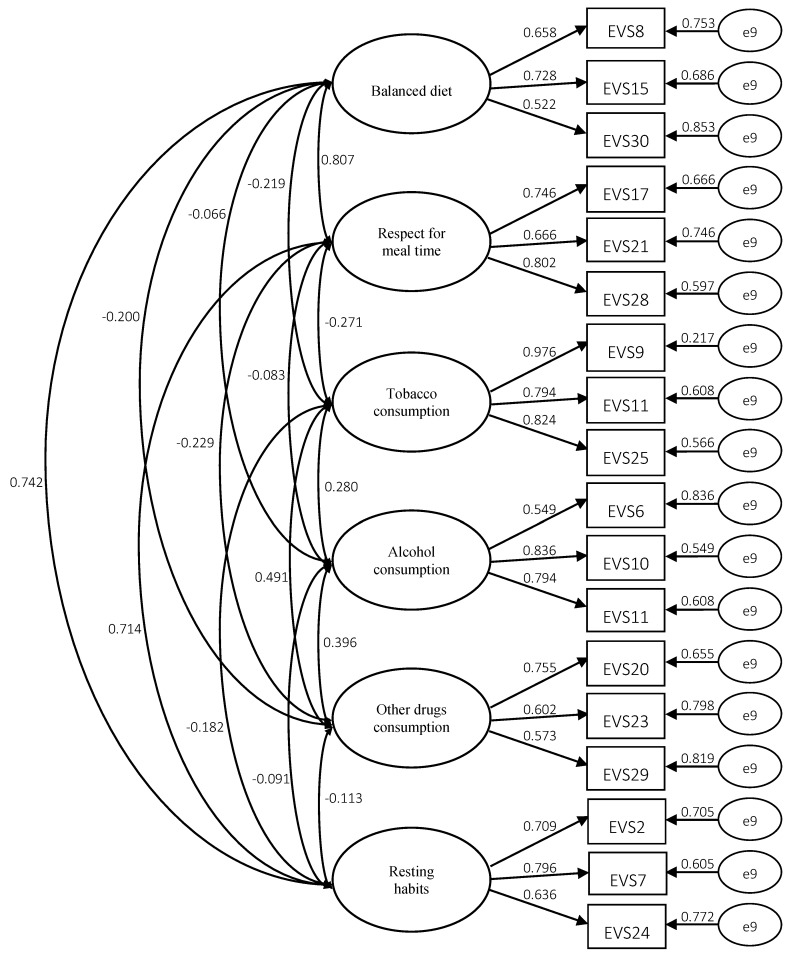
Structural model of the adapted version of the Healthy Lifestyle Questionnaire (EVS II).

**Table 1 ijerph-17-01458-t001:** Internal consistency values of EVS II.

Variable	Item	FL	CF	Ω	MVE	CR	OHS
Balanced diet	EVS 8	0.741	0.837 *	0.81	0.82	0.53	0.27
	EVS 15	0.707	0.658 *				
	EVS 19	0.582	0.530 *				
	EVS 26	0.639	0.630 *				
	EVS 30	0.708	0.762 *				
Respect for mealtime	EVS 4	0.699	0.682 *	0.90	0.91	0.71	0.36
	EVS 17	0.787	0.852 *				
	EVS 21	0.718	0.761 *				
	EVS 27	0.728	0.754 *				
	EVS 28	0.851	0.975 *				
Tobacco consumption	EVS 1	0.847	0.899 *	0.96	0.96	0.85	0.49
	EVS 9	0.944	0.986 *				
	EVS 14	0.865	0.848 *				
	EVS 25	0.903	0.980 *				
Alcohol consumption	EVS 6	0.700	0.539 *	0.76	0.79	0.51	0.35
	EVS 10	0.827	0.940 *				
	EVS 11	0.825	0.847 *				
Other drugs consumption	EVS 18	0.681	0.618 *	0.78	0.78	0.51	0.27
	EVS 20	0.776	0.742 *				
	EVS 23	0.762	0.767 *				
	EVS 29	0.685	0.625 *				
Resting habits	EVS 2	0.828	0.823 *	0.74	0.77	0.54	0.34
	EVS 7	0.875	0.982 *				
	EVS 24	0.732	0.560 *				

FL—Factor Loading: Correlation between item and factor; CF—Factorial load of the item in the factor * *p* < 0.01; Ω—McDonald’s Omega; MVE—Mean Variance Extracted; CR—Composite Reliability; OMS—Omega Hierarchical Subscale.

**Table 2 ijerph-17-01458-t002:** Descriptive statistics and concurrent validity between the variables of EVS II and BRSQ.

Variable	Mean	SD	1	2	3	4	5	6	7	8	9
1. Balanced diet	3.63	0.80	-	0.61 ^**^	−0.21 ^**^	−0.49	−0.13 ^*^	0.53 ^**^	0.14 ^*^	−0.28 ^**^	−0.34 ^**^
2. Respect for mealtime	3.64	0.89		-	−0.27 ^**^	−0.17 ^**^	−0.23 ^**^	0.61 ^**^	0.45 ^**^	−0.22 ^**^	−0.09
3.Tobacco consumption	1.57	1.02			-	0.33 ^**^	0.42 ^**^	−0.17 ^**^	−0.14 ^*^	0.20 ^**^	0.13 ^*^
4.Alcohol consumption	1.52	0.68				-	0.41 ^**^	−0.08	-0.11 ^*^	0.11 ^*^	0.05
5.Other drugs consumption	1.55	0.74					-	−0.12 ^*^	−0.01	0.18 ^**^	0.11 ^*^
6.Resting habits	3.40	0.93						-	0.10 ^*^	−0.11 ^*^	−0.11 ^*^
7. Autonomous motivation	5.43	0.89							-	−0.21 ^**^	−0.31 ^**^
8. Controlled motivation	1.99	1.07								-	0.76 ^**^
9.Amotivation	2.02	1.20									-

*Note*: * *p* < 0.05; ** *p* < 0.01.

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
