# Peer review of "Validation of the Portuguese Version of the Healthy Lifestyle Questionnaire"

_ijerph, 2020, doi:10.3390/ijerph17041458_

Round 1

Reviewer 1 Report

This research article addresses a relevant proposal for the advancement of science in the field of psychometrics and, in particular, in the field of health. My comments are intended to help authors publish a high quality article. I congratulate you and invite you to accept my recommendations in your article.

Summary:
• I recommend removing the statistical values ​​from the summary. Specifically, I recommend eliminating the following: x2 = 305.925, p = .000, fd = 120.017, œá2 / fd = 2.549, NFI = .909, TLI = .918, CFI = .944, IFI = .946, MFI = .901 , GFI = .944, AGFI = .909, RMR = .051, SRMR = .048, RMSEA = .060)

Introduction:
• I congratulate you on the synthesis of the subject, which is a complex one. I really liked the introduction. It gives context of why the phenomenon occurs socially and clarifies and educates the reader in a clear, simple and practical way.

Method:
• Research design: For greater methodological rigor, the design must be specific and include the scope of the study. Several authors (Ato, López & Benavente, 2013; Montero & León, 2007) have proposed different classification systems for research designs in psychology. I suggest the use of one of these systems for design classification. The manuscripts are:

Ato, M., López, J. J., & Benavente, A. (2013). A system of classification of research designs in psychology. Annals of Psychology, 29 (3), 1038-1059. Montero, I., & León, O. G. (2007). A guide for naming research studies in psychology. International Journal of Clinical and Health Psychology, 7 (3), 847-862.

Participants: I recommend that a greater description of the sample be made with specific data. The authors were very general in this section. Perhaps, they should place a table with the sociodemographic data of the sample.

Data Analysis: They made a good description of the analysis strategy. I only recommend that you divide the information into two paragraphs. The paragraph they have is very extensive.

Results:
• As a general recommendation and as a good practice when evaluating psychometric properties using exploratory and confirmatory factor analysis, it is not advisable to do the analysis using the same sample. These should be done with different samples. If the sample were large, I would recommend that they divide it in order to perform both analyzes. But, since the sample is considerably small, I recommend that the authors use a single method and perform the analyzes again. If they decide to do only the exploratory, they must eliminate the confirmatory analyzes of the manuscript. On the other hand, if they wish to carry out the CFA, they must begin the process again.

Discussion:
• The discussion should be arranged in the light of the authors' decision on which factor analysis they will perform.

Author Response

Dear Reviewer, I begin by thanking you for the opportunity to review the submitted article.

In view of the precious comments made on the document that we submitted to the International Journal of Environmental Research and Public Health, as authors we sought to respond to the overall suggestions, combining the assessments made by the reviewers.

From the point of view of drafting the text, the document was revised by a native teacher belonging to our working group, who suggested some changes.
All changes and new text extracts are marked in red.

The suggestion to eliminate the incremental indices of the introduction could be made as advised. However, when having to combine with other reviewers, they point to its maintenance, but only the most relevant indexes in the literature.
The introduction, although long, was maintained because it was considered in its very complete and pedagogical review regarding the theme of instruments to assess lifestyles.

We eliminated the suggested exploratory factor analysis, maintaining only the
confirmatory factor analysis of the instrument.
We consulted the proposed methodological works, based on the suggested authors, such as Montero & Leon (2007), making our manuscript more current and detailed.

The sample was described in a little more detail according to the request made.
The results were exposed in more paragraphs as requested as well.
We reduced the discussion of the results as requested, making it more specific as to the objective of the study, the confirmatory factor analysis.
In addition to your recommendations, the following changes were described below.

We also proceeded with the calculation of OmegaHs as suggested by one of the
reviewers and rectified the aspects mentioned regarding the dimensions of the
instrument's validity.

We eliminated table 3, as suggested by one of the reviewers, as it is not considered relevant in terms of the information provided to the reader.
We therefore hope to respond to comments and suggestions made regarding our
manuscript.

It would be a pleasure and of great importance to be able to publish our manuscript in the International Journal of Environmental Research and Public Health.

Best regards,
Marco Batista

Reviewer 2 Report

The manuscript describes the psychometric evaluation of the Portuguese version of the EVS II. In general, this is an interesting topic, but there are serious issues regarding language and statistical methods. Below are only the most important issues. When these issues are resolved, I will be happy to give a more comprehensive review.

(1) The wording of some sentences or whole sections is very difficult to understand. My recommendation is therefore to have a native speaker proofread the manuscript.

(2) The introduction and the discussion are far too long and not clear enough, i.e. it is not obvious why certain information is discussed. My recommendation is therefore to keep only the information in the manuscript that is important for the specific research question of the present study.

(3) Doing an exploratory factor analysis (EFA) and a confirmatory factor analysis (CFA) on the same data set makes no sense (Fokkema & Greiff, 2017). EFA is mainly intended to determine the number of factors. However, since the number of factors is already known (i.e. the items of the questionnaire are assigned to a certain subscale, e.g. Balanced Diet), CFA is the appropriate statistical method here. My recommendation: Remove EFA completely from the manuscript, as it does not provide any information that is not already included in CFA. (If necessary, the findings of EFA can be presented as a Supplement).

(4) Instead of Byrne (2006) I recommend using methodological articles (Schermelleh-Engel et al., 2003) as a reference for the cut-off values of the fit indices. The measurement models should then be evaluated using these cut-off values. By the way: Most of the fit indices are interdependent, so it is not useful to report all of them. Schermelleh-Engel et al. (2003) make recommendations as to which fit indices are actually useful.

(5) If  McDonald's Omega is reported, Cronbach's Alpha does not need to be reported. The requirements for Alpha are not met in this study (Dunn et al., 2013), so Omega is sufficient.

(6) Discriminant and convergent/concurrent validity usually means that different and identical constructs, respectively, are considered (Campbell & Fiske, 1959; Ziegler, 2014). This is not the case in this study, so the terms are not appropriate here.

(7) The analyses and the measurement model from Fig. 1 are not sufficient to check the factorial validity. The correlations between the factors are in some cases very high (i.e. r = .807), so it is very likely that some factors are not independent of each other. Gignac and Kretzschmar (2017) have presented a fairly simple approach how to check whether the number of factors is reasonable based on this specific measurement model (see Fig. 1). My recommendation is therefore to extend the analyses according to Gignac and Kretzschmar (2017).

(8) Table 8 provides virtually no gain in knowledge, since the fit indices of different questionnaires / measurement models can only be compared with each other to a limited extent. My recommendation: Delete the table.

*References*

Campbell, D. T., & Fiske, D. W. (1959). Convergent and Discriminant Validation by the Multitrait Multimethod Matrix. Psychological Bulletin, 56(2), 81–105. http://dx.doi.org/10.1037/h0046016

Dunn, T. J., Baguley, T., & Brunsden, V. (2013). From alpha to omega: A practical solution to the pervasive problem of internal consistency estimation. British Journal of Psychology, 1–14. https://doi.org/10.1111/bjop.12046

Fokkema, M., & Greiff, S. (2017). How Performing PCA and CFA on the Same Data Equals Trouble: Overfitting in the Assessment of Internal Structure and Some Editorial Thoughts on It. European Journal of Psychological Assessment, 33(6), 399–402. https://doi.org/10.1027/1015-5759/a000460

Gignac, G. E., & Kretzschmar, A. (2017). Evaluating dimensional distinctness with correlated-factor models: Limitations and suggestions. Intelligence, 62, 138–147. https://doi.org/10.1016/j.intell.2017.04.001

Schermelleh-Engel, K., Moosbrugger, H., & Müller, H. (2003). Evaluating the fit of structural equation models: Tests of significance and descriptive goodness-of-fit measures. Methods of Psychological Research Online, 8(2), 23–74.

Ziegler, M. (2014). Stop and State Your Intentions!: Let’s Not Forget the ABC of Test Construction. European Journal of Psychological Assessment, 30(4), 239–242. https://doi.org/10.1027/1015-5759/a000228

Author Response

Dear Reviewer, I begin by thanking you for the opportunity to review the submitted article.

In view of the precious comments made on the document that we submitted to the International Journal of Environmental Research and Public Health, as authors we sought to respond to the overall suggestions, combining the assessments made by the reviewers.

From the point of view of drafting the text, the document was revised by a native teacher belonging to our working group, who suggested some changes.
All changes and new text extracts are marked in red.

The introduction, although long, was maintained because it was considered by one of the reviewers to be quite complete and pedagogical as to the theme of the instruments to assess lifestyles. As we believe that it does not harm the structure of the article, we have chosen to keep it.

We eliminated the suggested exploratory factor analysis, maintaining only the
confirmatory factor analysis of the instrument, as suggested.
We proceeded to consult the proposed methodological works, based on the suggested authors, such as Schermelleh-Engel et al. (2003) or Gignac and Kretzschmar (2017), making our manuscript more current, detailed and scientifically accurate.

We also proceeded with the calculation of OmegaHs as suggested by one of the
reviewers and rectified the aspects mentioned regarding the dimensions of the
instrument's validity.

We eliminated table 3, as well as the suggestion made, as it is not considered relevant in terms of the information provided to the reader.

We reduced the discussion of the results as requested, making it more specific as to the study objective, the confirmatory factor analysis.

We therefore hope to respond to comments and suggestions made regarding our
manuscript.
It would be a pleasure and of great importance to be able to publish our manuscript in the International Journal of Environmental Research and Public Health.

Best regards,
Marco Batista

Round 2

Reviewer 1 Report

Accept in present form

Author Response

Dear Reviewer, I am grateful for the appreciation made for our article and for the fact that I have given a positive opinion.

Best regards,
Marco Batista

Reviewer 2 Report

I think the authors have done a relatively good job in revising the manuscript. However, there are still some issues that need to be improved:

(1) I strongly recommend having a native speaker proofread the manuscript. There are still phrases in the manuscript which are unusual or sometimes difficult to understand. There are also a few typos. Just a few examples (not complete!)
- Abstract, lines 25-28
- page 3, lines 144-146
- p5, l 243: Chi2/fd -> df (several times in the manuscript)

(2) Introduction (until 122): What is the benefit of presenting the different measuring instruments? The context is missing. In other words, what should the reader do with this information? Why is this information relevant to the study? It may well be that you want to give an overview of the different measuring instruments, but then you need an explanation. And the text should be better structured.

(3) Introduction (lines 130-138): This information seems completely irrelevant to the study. What is the purpose of listing the programmes? Only the sentence in lines 138 and 139 is important and relevant (i.e. study objective).

(4) Instruments: The BRSQ has 24 items / 6 subscales. Why are only 3 subscales reported in Table 2? If it is common practice to summarize different subscales, this should be explained in more detail.

(5) Data Analysis, line 209: As expl. factor analysis is no longer reported, the methods section should be adapted accordingly.

(6) Data Analysis: What about missing data? If there were no missing data, this should be mentioned. If there were missing data, then it should be explained how this was handled.

(7) Results: The order in which results are presented should be reconsidered. Reliability calculations only make sense if the factorial validity has been checked. And even convergent validities depend on the underlying measurement model. My suggestion would therefore be: First check the factorial validity, then present reliabilities and then convergent validities.

(8) Results, convergent validity: Are there any ad-hoc expectations or hypotheses on how the scales of the BRSQ should correlate with the scales of EVSII? Otherwise, the correlations in Table 2 are purely exploratory and can only be interpreted in a very limited way. My criticism from the last review, that the use of the terms convergent and discriminant validity only makes sense if there are ad-hoc expectations, is still relevant.

(9) Figure 1: In the methods section it was described that the EVSII would have 30 items. However, in the measurement model (Fig. 1) only 18 indicators of the latent variables are shown. If the indicators represent the items: Why are 12 items missing?

(10) Discussion, line 360-…: The information that items have been eliminated should already be shown in the results section. This is important for the interpretation of the results: Figure 1 is not the measurement model of the EVSII, but of an adapted version the EVSII - this is an important difference. Relevant in this sense is also: To what extent is the content validity of the subscales still comparable when items have been eliminated?

Author Response

Review 2

Dear Reviewer, I begin by thanking you for the opportunity to review the submitted article again.

In view of the precious comments made on the document that we submitted to the International Journal of Environmental Research and Public Health, as authors we tried to respond to the overall suggestions.

Regarding the first point of the recommendations, we changed the phrases marked by the reviewer, as well as the errors noted. From the point of view of drafting the text, the document was revised by a native teacher belonging to our working group, who suggested some changes. We hope that this version will present the necessary clarity required for a manuscript of this nature.

With regard to the second point of the recommendations, we tried to comply with the recommendations, namely to reduce the introduction and make it more specific, regarding the objective of the study.

In the third point of the recommendations, we deleted the irrelevant information highlighted by the reviewer.

In the fourth point of the recommendations, we tried to detail in the text the theoretical relationship between the motivation continuum, underlying the Theory of Self-Determination, with the consequence of behavioral variables, which in the present study, are the lifestyles.

Regarding the fifth point of the recommendations, the text suggested by the reviewer was adapted.

In the sixth point of the recommendations, we introduced in the text that there was no absence of data.

In point seven and other of the recommendations, the order of exposure of the data was changed according to the suggestions made.

Point nine and ten of the recommendations, we tried to explain the elimination of the items, introducing an explanation in the part of the results as suggested and keeping a shorter paragraph in the discussion.

Best regards,
Marco Batista